# Estimating Correlation Clustering Cost in Node-Arrival Stream

**Kaiwen Liu** [†1]  **Seba Daniela Villalobos** [†1]  **Qin Zhang** [†1]

## Abstract

We study the correlation clustering problem in the node-arrival data stream model. Unlike previous work, where the stream consists of the graph's edges, we focus on the setting in which the stream contains only the *nodes*. This model better reflects many real-world scenarios in which the data stream naturally consists of raw objects (e.g., images, tweets), and the similar/dissimilar edges are derived through a similarity function. We present $\texttt{C}^4\texttt{Approx}$, a streaming algorithm that approximates the cost of correlation clustering using sublinear space in the number of nodes and a constant number of passes. We further complement this result with lower bounds. Experiments on real-world datasets show that by storing only 2% of the nodes, our algorithm achieves performance comparable to the classic `Pivot` algorithm and the more recent `PrunedPivot` algorithm, even on sparse graphs.

## 1. Introduction

In the correlation clustering problem, we are given a complete graph with $\pm 1$ labels on the edges, where a '+1' (positive) edge denotes a pair of *similar* objects and a '−1' (negative) edge denotes a pair of *dissimilar* objects. The goal is to partition the nodes into clusters so as to minimize the total number of mismatches. A mismatch occurs in two cases: (1) a positive edge connects nodes in different clusters, or (2) a negative edge connects nodes in the same cluster.

Correlation clustering is a fundamental problem in computer science with a wide range of applications, including data deduplication (Arasu et al., 2009), community detection (Shi et al., 2021; Veldt et al., 2020), computer vision (Kim et al., 2011), natural language processing (Elsner & Schudy, 2009), bioinformatics (Bhattacharya & De, 2010), social network analysis (Li et al., 2017), and portfolio diversification (Zhan et al., 2015). The problem is NP-hard and even APX-hard (Bansal et al., 2004; Charikar et al., 2005). A number of $O(1)$-approximation algorithms have been proposed for correlation clustering in the past several decades (Bansal et al., 2004; Charikar et al., 2005; Ailon et al., 2008; Chawla et al., 2015; Cohen-Addad et al., 2022; 2023; Cao et al., 2024a; 2025). However, most of these algorithms are based on mathematical programming, which makes them difficult to implement in *big data* models such as the data stream model and distributed/parallel computation models.

For big data models, several pass- and round-efficient $O(1)$-approximation algorithms have been designed (Chierichetti et al., 2014; Ahn et al., 2015; Pan et al., 2015; Cohen-Addad et al., 2021; Behnezhad et al., 2022; Assadi & Wang, 2022; Behnezhad et al., 2023; Makarychev & Chakrabarty, 2023; Assadi et al., 2023; Cao et al., 2024b; Cambus et al., 2024; Cohen-Addad et al., 2024b; Dalirrooyfard et al., 2024), most of which build on the celebrated `Pivot` algorithm of Ailon et al. (2008), as will be discussed in details in Section 2.

**Node-Arrival Streams.** In this paper, we work on the data stream model (Alon et al., 1999), where the input arrives as a sequence of items over time. The algorithm may perform one or several sequential scans of the input stream and is required to compute a function of the stream's items at the end. The primary goal is to minimize the algorithm's space complexity.

Several works studied the correlation clustering problem in the data stream model (Ahn et al., 2015; Assadi & Wang, 2022; Behnezhad et al., 2023; Makarychev & Chakrabarty, 2023; Cambus et al., 2024), where the stream consists of the edges of the graph. We refer to this model as the *edge-arrival* stream model. These algorithms achieve space complexity of $O(n \operatorname{polylog}(n))$ words, where $n$ is the number of nodes in the graph.[1]

In many real-world applications, it is more natural to model the stream as containing the graph's nodes, while edge labels (+1 for similar and −1 for dissimilar) can be computed *on-demand* by a pairwise similarity function whenever both end nodes are in memory, rather than being stored explicitly.

---

†Authors listed in alphabetical order. [1]Department of Computer Science, Indiana University, Bloomington, IN, USA. Correspondence to: Qin Zhang <qzhangcs@iu.edu>.

*Proceedings of the 43rd International Conference on Machine Learning*, Seoul, South Korea. PMLR 306, 2026. Copyright 2026 by the author(s).

[1]We assume each node or edge can be stored in one word.

After all, the nodes correspond to the actual data objects (e.g., images, tweets), while the edges are derived quantities that encode similarity between nodes.

In node-arrival streams, a natural question is whether correlation clustering can be solved using space sublinear in the number of nodes $n$. With $O(n)$ space, the problem is trivial, since we can store all nodes and reconstruct the edges on-demand using the similarity function. However, if the goal is to output the actual clustering, $\Omega(n)$ space is unavoidable, as the number of clusters can be as large as $n$. In this work, rather than outputting a full clustering, we instead aim to learn the "clusterability" of the input graph. Specifically, we focus on approximating the *cost* of correlation clustering, defined as the number of mismatch pairs. This paper addresses the following question:

*Can we obtain a good approximation of the cost of optimal correlation clustering in node-arrival streams using $o(n)$ space?*

**Motivations.** We give two concrete applications for estimating the cost of correlation clustering in node-arrival streams.

*Quantifying data inconsistency.* The first application is to measure the inconsistency in a dataset $\sigma = (\sigma_1, \ldots, \sigma_n)$. Imperfect data acquisition, mismatches in how information is encoded, and distortions introduced during processing can blur true relationships: items that represent different underlying entities may appear similar, while those from the same entity may seem dissimilar. When representing the dataset $\sigma$ as a signed complete graph $G^\sigma$, where each item corresponds to a node, with similar pairs assigned edges labeled '+1' and dissimilar pairs assigned edges labeled '−1', the minimum number of inconsistent pairs is exactly the correlation clustering cost. A recent paper (Liu & Zhang, 2026) introduces the concept of $F_p$-*mismatch-ambiguity* to quantify the impact of data inconsistency on the $F_p$ problem. For $p = 2$, $F_2$-mismatch-ambiguity is essentially the correlation clustering cost. While this parameter appears in the approximation guarantees of streaming algorithms for the frequency moment $F_2$, their work does not provide a method for estimating it in the streaming model. Our work is therefore complementary, as it focuses on approximating this quantity in the data stream model.

*Learning a good similarity function.* Another application is selecting an appropriate similarity function, parameterized by $\theta$, so that the dataset becomes well clusterable. The correlation clustering cost induced by this function provides a natural measure for evaluating its effectiveness. For instance, in our experiments on the ImageNet-21K dataset, each item is embedded into a vector in $\mathbb{R}^{1024}$, and similarity is determined using a threshold $\theta$ on cosine similarity. The parameter $\theta$ can be optimized via grid search to minimize the resulting clustering cost. Streaming algorithms enable efficient estimation of this cost through linear scans over the data while using only a small memory.

**Our Contribution.** We say $\tilde{X}$ is a $(c, d)$-approximation of $X$ if $X \leq \tilde{X} \leq cX + d$; when $d = 0$, we simply say $\tilde{X}$ is a $c$-approximation of $X$. We say $\tilde{X}$ is a $(1 \pm \epsilon, \pm d)$-approximation of $X$ if $(1 - \epsilon)X - d \leq \tilde{X} \leq (1 + \epsilon)X + d$. Our contributions are as follows.

1. Our main result is the algorithm $\texttt{C}^4\texttt{Approx}$ (*Compact Correlation Clustering Cost Approximator*) that gives the following guarantee:

   - For any constant $\kappa \in (0, 1]$, there exists a streaming algorithm in the node-arrival model that computes a $(O(1), n^\kappa)$-approximation to the cost of optimal correlation clustering on an $n$-node graph with probability 0.99. The algorithm runs in $O(1)$ passes and uses $O(n^{1-\kappa/4} \log n)$ words of space.

2. We complement this algorithmic result with two lower bounds, which justify the use of both relative and additive errors in algorithms with $O(1)$-pass complexity.

   - *Multiple passes are necessary:* for any $c \geq 1$ and $d \in (0, \frac{n}{3}]$, any one-pass 0.49-error $(c, d)$-approximation streaming algorithm for computing the cost of the optimal correlation clustering on an $n$-node graph in the node-arrival model needs $\Omega(n)$ bits of space.

   - *Additive error is unavoidable:* for any $c \geq 1$, any $O(1)$-pass 0.49-error $(c, 0)$-approximation streaming algorithm for computing the cost of the optimal correlation clustering on an $n$-node graph in the node-arrival model needs $\Omega(n)$ bits of space.

3. We evaluate $\texttt{C}^4\texttt{Approx}$ using real-world datasets. The results show that $\texttt{C}^4\texttt{Approx}$ achieves clustering cost comparable to $\texttt{Pivot}$ while requiring significantly less memory.

**A Comparison with Assadi et al. (2023).** In the edge-arrival model, Theorem 1 of (Assadi et al., 2023) shows that there is an algorithm that achieves an $(O(1), \delta n^2)$-approximation using $\text{polylog}(n)/\delta^5$ words of space, which is polylogarithmic only when $\delta$ is a constant. Their algorithm can be implemented in the node-arrival model with the same space and approximation guarantees. In contrast, our algorithm achieves an $(O(1), n^\kappa)$-approximation using $O(n^{1-\kappa/4} \log n)$ space for any $\kappa > 0$. For example, setting $\kappa = 0.1$ yields an $(O(1), n^{0.1})$-approximation with still sublinear space. Matching this additive error in (Assadi et al.,

2023) would require setting $\delta = n^{-1.9}$, which in turn gives a space complexity of $\text{polylog}(n)/\delta^5 = \Omega(n^{9.5})$.

Generally speaking, node-arrival and edge-arrival streams are not directly comparable. In node-arrival streams, edges can only be queried when both endpoints are simultaneously stored, making it impossible to even enumerate all edges in $o(n)$ space using constant passes. In contrast, edge-arrival streams allow a full scan of edges in each pass. On the other hand, the node-arrival model permits querying edges between stored nodes at any time, which may require additional passes in the edge-arrival setting.

In Section 4, we have empirically compared our main algorithm $\mathtt{C^4Approx}$ with the two algorithms proposed in (Assadi et al., 2023). Our results demonstrate that $\mathtt{C^4Approx}$ clearly outperforms both algorithms in (Assadi et al., 2023) in terms of stability and accuracy, especially on sparse graphs.

**Other Related Work.** (Behnezhad et al., 2025; 2019; Chechik & Zhang, 2019; Cohen-Addad et al., 2024a; Dalirrooyfard et al., 2025) study correlation clustering in dynamic settings, considering various update models including edge insertions and deletions as well as node insertions and deletions. Their main focus is on update time, and their algorithms still require at least linear space.

## 2. The Algorithm

**The Pivot Algorithm.** We start by reviewing the $\mathtt{Pivot}$ algorithm (Ailon et al., 2008) and the $\mathtt{PrunedPivot}$ algorithm (Dalirrooyfard et al., 2024). Let $V$ be the set of $n$ input nodes, and let $G$ be the undirected graph on $V$ induced by the similarity function, containing only positive edges.

In $\mathtt{Pivot}$, we first choose a random permutation $\pi$ of $V$, where $\pi(u)$ denotes the rank of node $u$. A node $u$ is said to have a *higher* rank than $v$ if $\pi(u) < \pi(v)$. The algorithm iteratively selects the highest ranked remaining node as the *pivot*, forms a cluster consisting of the pivot and its remaining neighbors, and removes these nodes from the graph. This simple algorithm achieves a 3-approximation of the cost of the optimal correlation clustering.

The $\mathtt{Pivot}$ algorithm can be formulated recursively. Let $\text{pivot}_\pi(u)$ be the pivot node of the cluster containing $u$ w.r.t. permutation $\pi$, and $N(u)$ be the set of neighbors of $u$. It suffices to determine $\text{pivot}_\pi(u)$ for each $u \in V$. Note that $\text{pivot}_\pi(u)$ is either $u$ or some $v \in N(u)$ with a higher rank. Thus, computing $\text{pivot}_\pi(u)$ reduces to computing $\text{pivot}_\pi(v)$ for each neighbor $v \in N(u)$ with $\pi(v) < \pi(u)$. If no such neighbor is a pivot, $u$ itself serves as the pivot.

The $\mathtt{PrunedPivot}$ algorithm adds an early-stopping rule to the recursive formulation of the $\mathtt{Pivot}$ algorithm: if

*Table 1.* Summary of notations

| Notation | Meaning |
| --- | --- |
| $V$ | input stream $(\sigma_1, \ldots, \sigma_n)$ |
| $G$ | positive-edge graph induced by similarity |
| $N(u)$ | neighbors of $u$ |
| $d_u$ | degree of $u$ in $G$ |
| $\pi(u)$ | rank of $u$ in $\pi$ |
| $\text{pivot}_\pi(u)$ | pivot of $u$ under $\mathtt{PrunedPivot}$ |
| $R$ | highest-ranked stored nodes |
| $A, B$ | pivot determined / not determined by $R$ |
| $k$ | pivot-search depth limit |
| $E_\pi^{\text{mis}}$ | mismatch pairs under $\pi$ |
| $\alpha$ | constant controls space-error tradeoffs |
| $\beta$ | $\beta = (1-\alpha)/4$ |

$\text{pivot}_\pi(u)$ remains unidentified after $k$ recursive calls for a fixed parameter $k$, the recursion terminates and $u$ is placed in a singleton cluster. This modification renders $\mathtt{PrunedPivot}$ suitable for parallel and local computation models.

**Theorem 2.1** (Dalirrooyfard et al. (2024)). *With probability at least $2/3$, the cost of the clustering produced by $\mathtt{PrunedPivot}$ with parameter $k$ is a $\left(9 + \frac{24}{k-1}\right)$-approximation of the cost of the optimal clustering.*

To facilitate discussion, we introduce the following notions. We say $u \sim v$ $(u \nsim v)$ if there is a $+1$ $(-1)$ edge between node $u$ and node $v$. For convenience, we also say $u \sim u$ for any $u \in V$.

**Definition 2.2** (mismatch pair). *Given a permutation $\pi$, a pair of nodes $(u, v)$ is called a mismatch pair if $u \sim v$ but $\text{pivot}_\pi(u) \neq \text{pivot}_\pi(v)$, or if $u \nsim v$ but $\text{pivot}_\pi(u) = \text{pivot}_\pi(v)$.*

**Definition 2.3** (mismatch graph). *Given a permutation $\pi$, let $G_\pi^{mis} = (V, E_\pi^{mis})$ denote the mismatch graph of $G$ under $\pi$, where the edge set $E_\pi^{mis}$ consists of all mismatch pairs in $V \times V$ under $\pi$.*

For convenience, when $\pi$ is clear from context, we will abbreviate $\text{pivot}_\pi(\cdot), G_\pi^{\text{mis}}, E_\pi^{\text{mis}}$ as $\text{pivot}(\cdot), G^{\text{mis}}, E^{\text{mis}}$. We will use a constant parameter $\alpha \in [0, 1)$ to control the tradeoffs between the (additive) approximation factor and the space usage of our algorithm. We provide a list of key notations in Table 1.

**High-Level Ideas of $\mathtt{C^4Approx}$.** Before presenting our algorithm, we try to illustrate the main ideas. Our goal is to estimate $\left| E^{\text{mis}} \right|$, the cost of the clustering produced by $\mathtt{PrunedPivot}$, which by Theorem 2.1 is an $O(1)$-approximation of the optimal clustering cost with a good probability for any $k \geq 2$. Note that for each node $u$,

pivot($u$) can be computed by executing at most $k$ recursive calls. This gives a naive $O(k)$-pass algorithm (referred to as `SimpleSampling` hereafter): we sample $q$ pairs of nodes, find their pivots and test whether they belong to $E^{\text{mis}}$ according to Definition 2.2, and then scale back. However, this straightforward approach incurs a large additive error $\Theta(n^2/\sqrt{q})$, which is $\omega(n^{1.5})$ when $q = o(n)$.

We hope to further improve `SimpleSampling`, especially by reducing its additive error. The main difficulty is that under an $o(n)$ space budget, we cannot store the pivot information for all nodes. Consequently, we do not have an oracle for $E^{\text{mis}}$, which, given two nodes, determines whether $(u, v) \in E^{\text{mis}}$ *directly* without going through the $O(k)$-pass recursive search.

Our key idea is that we can make a *partial oracle* for $E^{\text{mis}}$ by maintaining a small reference set $R$ in memory, consisting of the $\tilde{O}(n^{1-\beta})$ highest ranked nodes according to the random permutation $\pi$. This allows us to directly compute pivot($u$) for all nodes $u$ with pivot($u$) $\in R$. Moreover, even if pivot($u$) $\notin R$, pivot($u$) can still be computed directly as long as the degree of $u$ in $G$ is high. The reason is that pivot($u$) depends on its $k$ highest ranked neighbors, which are contained in $R$ with high probability if $u$ is a high degree node.

Let $A$ be the set of nodes whose pivots can be directly determined by $R$, and let $B = V \setminus A$. It is easy to show that all nodes in $B$ are low-degree nodes with high probability. We can further show that if the clusters are generated by running `PrunedPivot` on $G$, then each cluster lies entirely within either $A$ or $B$. We thus divide $E^{\text{mis}}$ into two parts that can be estimated separately.

The first part is $E_A^{\text{mis}}$, which is the set of edges with *at least one* end node in $A$. We can show that for any two nodes $u$ and $v$, if either $u \in A$ or $v \in A$, then the reference set $R$ together with the similarity function suffices to determine whether $(u, v)$ is a mismatch edge. The problem then reduces to estimating the average degree of the subgraph $G_A^{\text{mis}} = (V, E_A^{\text{mis}})$ in the node-arrival stream given access to an edge query oracle. Simple subsampling does *not* work for this task, as node degrees range over $\{0, 1, \ldots, n-1\}$, leading to prohibitively high variance. Instead, we partition the set $V$ into $H$ and $L$, the high and low-degree nodes in $G_A^{\text{mis}}$ respectively. Such a partition can be done by sampling a subset $S_1$. For each node $u$, let $N_A^{\text{mis}}(u)$ be its neighborhood in $G_A^{\text{mis}}$. Then, $\left| N_A^{\text{mis}}(u) \cap S_1 \right|$ acts as a certificate for determining if $u$ is a high-degree node, and if it is, we can estimate its degree through appropriate rescaling. For the remaining (low-degree) nodes, we can just use subsampling, as the estimator exhibits low variance and therefore gives a good approximation.

The second part is $E_B^{\text{mis}}$, which consists of edges with *both*

end nodes in $B$. For two nodes $u, v \in B$, we cannot use $R$ to determine whether $(u, v) \in E^{\text{mis}}$. We thus take a different approach. Note that $B$ only contains low-degree nodes, so all clusters within $B$ are small. Let $\mathcal{C}(B)$ denote the set of clusters in $B$. We can directly sample from $\mathcal{C}(B)$ and store the sampled clusters entirely. We then count the intra- and inter-cluster edges incident to the sampled clusters, which can be rescaled to obtain an estimate of $\left| E_B^{\text{mis}} \right|$. As the contribution of each cluster is bounded, the variance of this estimator is small, yielding a good approximation.

Our algorithm is based on two key ideas: (1) a sublinear-size reference set for partitioning nodes, and (2) a high-low degree decomposition to control variance of the sampling-based estimations. Although developed for correlation clustering, these techniques may extend to other graph problems in node-arrival streams.

**Node Partition Using $R$.** We now present Algorithm 1 `FindPivot`, a variant of `PrunedPivot` (Dalirrooyfard et al., 2024) that restricts the search of pivot($u$) for a query node $u \in V$ within the set $R$. The node partition $V = A \uplus B$ is then determined by running `FindPivot` on each node in $V$.

---

**Algorithm 1** `FindPivot`$(u, \pi, R)$

**Input:** query node $u$, permutation $\pi : V \to [n]$, a subset $R \subseteq V$ of highest ranks
**Output:** pivot($u$) or `null`
1   Initialize a global variable $\gamma \leftarrow 0$
2   $p \leftarrow$ `Rec-FindPivot`$(u, \pi, R)$
3   **if** $p =$ `timeout` **then return** $u$
4   **return** $p$

---

**Procedure** `Rec-FindPivot`$(u, \pi, R)$
5     **if** $\gamma \geq k$ **then return** `timeout`
6     $Q(u) \leftarrow \{v \in (N(u) \cup \{u\}) \cap R \mid \pi(v) \leq \pi(u)\}$
7     sort $Q(u)$ in ascending order w.r.t. $\pi$
8     **foreach** $v \in Q(u)$ **do**
9       **if** $v = u$ **then return** $u$
10      $\gamma \leftarrow \gamma + 1$
11      $p \leftarrow$ `Rec-FindPivot`$(v, \pi, R)$
12      **if** $p =$ `timeout` *or* $p = v$ **then return** $p$
13     **return** `null`

---

When $R = V$, `FindPivot` is identical to `PrunedPivot`. When $R \subset V$, the search processes of `FindPivot` and `PrunedPivot` coincide when the query is restricted to $R$. `FindPivot` terminates under one of the following conditions: (1) the pivot of $u$ is found; (2) the query budget $k$ is exhausted (i.e., `timeout`); (3) $u \notin R$, no neighbor of $u$ in $R$ is identified as a pivot, and the query budget has *not* been reached. In Case (1), since all recursive calls made by $u$ are restricted to nodes in $R$, we must have

$\text{pivot}(u) \in R$. In Case (2), $u$ is placed in a singleton cluster. In Case (3), $\text{pivot}(u)$ cannot be determined only using $R$, and `FindPivot` returns `null`. We note that `FindPivot` and `PrunedPivot` differ only in Case (3).

**Definition 2.4** ($(A, B)$ node partition w.r.t. $R$). *For a reference set $R$, let $A = \{u \in V \mid \text{FindPivot}(u, \pi, R) \neq \text{null}\}$ denote the set of nodes whose pivots can be determined by $R$. Let $B = V \backslash A$.*

In $\text{C}^4\text{Approx}$, once the set $R$ is identified, we store it in memory. Thus, `FindPivot` can be executed without any additional space or pass.

We will still use the original `PrunedPivot` to identify the pivots of nodes in the set $B$ on demand. `PrunedPivot` can be implemented in the data stream model using $k$ passes and $O(k)$ space. For completeness, we include a non-recursive streaming implementation in Appendix B.

The following two lemmas give the key properties of the $(A, B)$ partition. The first states that each cluster, as defined by the pivots of the nodes in $V$, consists entirely of nodes of the same type.

**Lemma 2.5.** *For any node $u$, we have $u \in A$ iff $\text{pivot}(u) \in A$, and $u \in B$ iff $\text{pivot}(u) \in B$.*

*Proof.* We can just prove $u \in A$ if and only if $\text{pivot}(u) \in A$; the second claim follows immediately.

If $u \in A$, we either have $\text{pivot}(u) \in R \subseteq A$, or $u$ is placed in a singleton cluster, which means $\text{pivot}(u) = u \in A$.

For the other direction, if $\text{pivot}(u) \in A$, we consider two cases. First, if $\text{pivot}(u) \in R$, then $u \in A$, and we are done. Otherwise, let $v = \text{pivot}(u) \in A \backslash R$. Since $v$ is a pivot, we have $\text{pivot}(v) = v \notin R$, which implies that $v$ forms a singleton cluster. Consequently, $u = v \in A$. $\square$

The next lemma shows that nodes in $B$ have small degrees, which will be useful for analyzing the variance of our estimator for their contribution to the clustering cost (see the proof of Lemma 2.9).

**Lemma 2.6.** *With probability at least $(1 - n^{-3})$, for every node $u \in B$, it holds that $d_u \leq n^\beta$.*

*Proof.* For any node $u$ with $d_u = |N(u)| > n^\beta$, $\mathbf{E}[|N(u) \cap R|] = \frac{r}{n} \cdot d_u > 48k \log n$. By a Chernoff bound (Lemma A.1),

$$\mathbf{Pr}[|N(u) \cap R| < k]$$
$$\leq \mathbf{Pr}\left[|N(u) \cap R| < \frac{1}{2}\mathbf{E}[|N(u) \cap R|]\right]$$
$$\leq \exp\left(-\frac{\mathbf{E}[|N(u) \cap R|]}{12}\right)$$
$$\leq n^{-4}.$$

By a union bound, with probability at least $(1 - n \cdot n^{-4}) = (1 - n^{-3})$, $|N(u) \cap R| \geq k$ for all nodes $u \in V$ with $d_u > n^\beta$. Running `FindPivot` on each such node $u$ triggers at most $k$ recursive calls in $R$, after which either $\text{pivot}(u)$ is found or the query budget $k$ is exhausted, in which case $u$ is placed in a singleton cluster. In both cases, $u \in A$. Therefore, any node $u \in B$ must have degree at most $n^\beta$. $\square$

**The Main Algorithm.** Our main algorithm is presented in Algorithm 2, which uses Algorithm 3 and 5 as subroutines.

---

**Algorithm 2** $\text{C}^4\text{Approx}(\sigma, \epsilon)$

**Input:** data stream $\sigma = (\sigma_1, \ldots, \sigma_n)$, parameter $\epsilon \in (0, 1)$
**Output:** an $(O(1), \epsilon n^{1-\alpha})$-approx of optimal correlation clustering cost on the induced graph of $\sigma$

14   $r \leftarrow 48kn^{1-\beta} \log n$, $R \leftarrow \emptyset$
15   $\pi \leftarrow$ a random permutation of $[n]$
    // Pass 1:
16   **foreach** *incoming node $\sigma_j$* **do**
17     |   **if** $\pi(\sigma_j) \leq r$ **then**   $R \leftarrow R \cup \{\sigma_j\}$
    // Pass 2 to $(4 + k)$:
18   **do in parallel**
19     |   $\tilde{m}_A \leftarrow \text{Est-EA}(\sigma, \epsilon/8, \pi, R)$
20     |   $\tilde{m}_B \leftarrow \text{Est-EB}(\sigma, \epsilon/8, \pi, R)$
21   **return** $(\tilde{m}_A + \tilde{m}_B + \frac{3}{8}\epsilon n^{1-\alpha})/(1 - \frac{\epsilon}{8})$

---

**Definition 2.7.** *For a fixed reference set $R$, let $(A, B)$ be the node partition of $V$ induced by $R$, and let $E^{mis}$ be the set of mismatch pairs (Definition 2.2). We partition $E^{mis}$ into two disjoint subsets $E_A^{mis}$ and $E_B^{mis}$, defined as*

$$E_A^{mis} = \{(u, v) \in E^{mis} \mid u \in A \text{ or } v \in A\} \quad \text{and}$$
$$E_B^{mis} = \{(u, v) \in E^{mis} \mid u \in B \text{ and } v \in B\}.$$

In Algorithm 2, we first fix $R$ as the set of the $48kn^{1-\beta} \log n$ highest ranked nodes under $\pi$. We then use Algorithm 3 to estimate $|E_A^{mis}|$ and Algorithm 5 to estimate $|E_B^{mis}|$; these two subroutines are discussed in the next two subsections. Recall that $|E_A^{mis}| + |E_B^{mis}|$ is the cost of the clustering using `PrunedPivot`, which is a $(9 + \frac{24}{k-1})$-approximation of the cost of the optimal clustering with probability $2/3$ by Theorem 2.1.

The following two lemmas summarize the performance of Algorithm 3 and 5, respectively. Due to space constraints, we leave the proof of Lemma 2.9 to Appendix C.1. The proof is similar to that for Lemma 2.8.

**Lemma 2.8.** *With probability $1 - o(1)$, Algorithm 3 outputs a $(1 \pm \epsilon, \pm \epsilon n^{1-\alpha})$-approximation of $|E_A^{mis}|$; it uses $O\left(\frac{1}{\epsilon^2}(n^{1-\beta} + n^{\alpha+\beta}) \log n\right)$ words of space and 3 passes.*

**Lemma 2.9.** *With probability $1 - o(1)$, Algorithm 5 outputs a $(1 \pm \epsilon, \pm \epsilon n^{1-\alpha})$-approximation of $|E_B^{mis}|$; it uses $O\left(\frac{k}{\epsilon^2} n^{\alpha+3\beta} \log n\right)$ words of space and $(k + 3)$ passes.*

These two lemmas directly lead to the following lemma; its proof can be found in Appendix C.2.

**Lemma 2.10.** *For any $\epsilon \in (0,1)$, with probability $1 - o(1)$, Algorithm 2 computes a $\left(1 + \frac{\epsilon}{3}, \epsilon n^{1-\alpha}\right)$-approximation of $\left|E^{mis}\right|$; it uses $(k + 4)$ passes and $O\left(\frac{k}{\epsilon^2}(n^{1-\beta} + n^{\alpha+3\beta}) \log n\right)$ words of space.*

Recall that we have chosen $\beta = \frac{1-\alpha}{4}$. Setting $k = 37$ and $\epsilon = \frac{1}{10}$, we get the following result, whose proof can be found in Appendix C.3.

**Theorem 2.11** (Main theorem). *For any constant $\alpha \in [0,1)$, there exists an algorithm that computes a $(O(1), n^{1-\alpha})$-approximation of the cost of optimal correlation clustering with probability $0.99$ for the induced graph of a data stream of length $n$ in the node-arrival model. It uses $O(1)$ passes and $O(n^{\frac{3+\alpha}{4}} \log n)$ words of space.*

### 2.1. Description of Algorithm 3 Est-EA

Let $G_A^{\mathrm{mis}} = (V, E_A^{\mathrm{mis}})$ be the subgraph of $G^{\mathrm{mis}}$ that only contains the edges from $E_A^{\mathrm{mis}}$. For each node $u$, let $N_A^{\mathrm{mis}}(u)$ be the set of neighbors of $u$ in $G_A^{\mathrm{mis}}$, and $d_A^{\mathrm{mis}}(u) = \left|N_A^{\mathrm{mis}}(u)\right|$ be the degree of $u$ in $G_A^{\mathrm{mis}}$. Clearly $\left|E_A^{\mathrm{mis}}\right| = \frac{1}{2}\sum_{u \in V} d_A^{\mathrm{mis}}(u)$.

In Algorithm 3, we first draw a sample $S_1$ of size $t_1$ from $V$ (Pass 1). Next, for each node $\sigma_j \in V$, we compute the intersection size between $N_A^{\mathrm{mis}}(\sigma_j)$ and $S_1$ (Pass 2, Line 26). We do this by calling a subroutine (Algorithm 4) on each $u \in S_1$ to determine if $(u, \sigma_j) \in E_A^{\mathrm{mis}}$. If the intersection is large enough, $d_A^{\mathrm{mis}}(\sigma_j)$ can be accurately estimated by rescaling the intersection size by $n/t_1$, and the estimate is then added to the sum (Line 27). Otherwise, $d_A^{\mathrm{mis}}(\sigma_j)$ must be small. Let $L$ be the set of such nodes. We draw another sample $S_2$ from $L$ (Pass 2, Line 29-33), and compute the degrees of nodes in $S_2$ (w.r.t. $E_A^{\mathrm{mis}}$) exactly (Pass 3, Line 36). The sum of the degrees of nodes in $S_2$, after rescaling, provides an accurate estimate of $\sum_{u \in L} d_A^{\mathrm{mis}}(u)$, because nodes in $L$ have small degrees. The final output equals one half of the sum of the contributions from high-degree nodes and low-degree nodes.

The following lemma shows that a sublinear size sample suffices to partition the nodes in $V$ to high and low degree sets w.r.t. $d_A^{\mathrm{mis}}(\cdot)$, which is needed for making the decision at Line 27 of Algorithm 3. Its proof is in Appendix C.4.

**Lemma 2.12.** *For any node $u \in V$, let $X_u$ denote $\left|N_A^{mis}(u) \cap S_1\right|$, and $t_1$ be defined at Line 22 of Algorithm 3. With probability at least $1 - O(n^{-4})$, the following holds: (1) if $X_u \geq \frac{2t_1}{n^{1-\beta}}$, then $(1-\epsilon)d_A^{mis}(u) \leq \frac{nX_u}{t_1} \leq (1+\epsilon)d_A^{mis}(u)$; (2) if $X_u < \frac{2t_1}{n^{1-\beta}}$, then $d_A^{mis}(u) \leq 4n^\beta$.*

The following lemma states that we can use Algorithm 4 to check whether a pair of nodes form an edge in $E_A^{\mathrm{mis}}$ when the reference set $R$ is stored, enabling Line 26 and Line 36

---

**Algorithm 3** Est-EA$(\sigma, \epsilon, \pi, R)$

**Input:** data stream $\sigma = (\sigma_1, \ldots, \sigma_n)$, parameter $\epsilon$, permutation $\pi : V \to [n]$, highest ranked nodes $R$

**Output:** a $(1 \pm \epsilon, \pm \epsilon n^{1-\alpha})$-approximation of $\left|E_A^{\mathrm{mis}}\right|$

22  $t_1 \leftarrow \frac{12}{\epsilon^2}n^{1-\beta}\log n$, $t_2 \leftarrow \frac{32}{\epsilon^2}n^{\alpha+\beta}\log n$, $n_\ell \leftarrow 0$, $\tilde{d}_A \leftarrow 0$
23  Initialize arrays $S_1[1..t_1] \leftarrow$ null, $S_2[1..t_2] \leftarrow$ null, and $D[1..t_2] \leftarrow 0$
    // Pass 1:
24  $S_1 \leftarrow$ Reservoir-sample $t_1$ nodes from $\sigma$
    // Pass 2:
25  **foreach** *incoming* $\sigma_j$ **do**
26  $\quad X_j \leftarrow \left|\{u \in S_1 \mid \text{In-EA}(u, \sigma_j, \pi, R)\}\right|$
27  $\quad$ **if** $X_j \geq \frac{2t_1}{n^{1-\beta}}$ **then** $\tilde{d}_A \leftarrow \tilde{d}_A + \frac{nX_j}{t_1}$
28  $\quad$ **else**
29  $\quad\quad n_\ell \leftarrow n_\ell + 1$
30  $\quad\quad$ **if** $n_\ell \leq t_2$ **then** $S_2[n_\ell] \leftarrow \sigma_j$
31  $\quad\quad$ **else**
32  $\quad\quad\quad i \leftarrow$ random integer from $[n_\ell]$
33  $\quad\quad\quad$ **if** $i \leq t_2$ **then** $S_2[i] \leftarrow \sigma_j$
    // Pass 3:
34  **foreach** *incoming* $\sigma_j$ **do**
35  $\quad$ **for** $i \leftarrow 1$ **to** $t_2$ **do**
36  $\quad\quad$ **if** $\text{In-EA}(S_2[i], \sigma_j, \pi, R)$ **then** $D[i] \leftarrow D[i] + 1$
37  $\tilde{d}_A \leftarrow \tilde{d}_A + \frac{n_\ell}{t_2}\sum_{i \in [t_2]} D[i]$
38  **return** $\tilde{d}_A/2$

---

in Algorithm 3. Its proof uses Lemma 2.5 and can be found in Appendix C.5.

**Lemma 2.13.** *Given the reference set $R$, for any two nodes $u, v \in V$, Algorithm 4 determines whether $(u, v) \in E_A^{mis}$.*

**Proof of Lemma 2.8.** With Lemma 2.12 and Lemma 2.13 in hand, we can now prove Lemma 2.8.

*Proof.* By Lemma 2.12 and a union bound, with probability $1 - O(n^{-4}) \cdot n = 1 - O(n^{-3})$, for every node $u$ with $X_u \geq \frac{2t_1}{n^{1-\beta}}$, we obtain a $(1 \pm \epsilon)$-approximation of $d_A^{\mathrm{mis}}(u)$ and add it to $\tilde{d}_A$ (Line 27 of Algorithm 3).

It remains to approximate the total degree of nodes in $L = \{u \in V \mid X_u < \frac{2t_1}{n^{1-\beta}}\}$. Again by Lemma 2.12 and a union bound, with probability $1 - O(n^{-3})$, every node $u \in L$ has degree $d_A^{\mathrm{mis}}(u) \leq 4n^\beta$.

Let $d_A^{\mathrm{mis}}(L) = \sum_{u \in L} d_A^{\mathrm{mis}}(u)$, and $t_2$ be defined at Line 22 of Algorithm 3. For each $i \in [t_2]$, let $Y_i$ be the $i$-th sample in $S_2$, and $Z_i = |L| \cdot d_A^{\mathrm{mis}}(Y_i)$. We have, $\mathbf{E}[Z_i] = |L|^{-1} \sum_{u \in L} |L| \cdot d_A^{\mathrm{mis}}(u) = d_A^{\mathrm{mis}}(L)$, and $Z_i \leq n \cdot (4n^\beta) = 4n^{1+\beta}$.

Let $\Delta = \max(d_A^{\mathrm{mis}}(L), n^{1-\alpha})$. By Bernstein's inequality

**Algorithm 4** In-EA$(u, v, \pi, R)$

**Input:** nodes $u, v$, permutation $\pi : V \to [n]$, subset $R \subseteq V$
**Output:** whether $(u, v) \in E_A^{\text{mis}}$
39  $p_u \leftarrow$ FindPivot$(u, \pi, R)$
40  $p_v \leftarrow$ FindPivot$(v, \pi, R)$
41  **if** $p_u = p_v =$ null **then**
42     |  **return** false
43  **else if** $p_u =$ null *or* $p_v =$ null **then**
44     |  **if** $u \sim v$ **then return** true
45  **else if** $(p_u = p_v) \wedge (u \nsim v)$ *or* $(p_u \neq p_v) \wedge (u \sim v)$ **then**
46     |  **return** true
47  **return** false

(see Lemma A.2),

$$\mathbf{Pr}\left[\left|\frac{1}{t_2}\sum_{i\in[t_2]}Z_i - d_A^{\text{mis}}(L)\right| \geq \epsilon\Delta\right]$$
$$\leq 2\exp\left(-\frac{t_2\epsilon^2\Delta^2}{4n^{1+\beta}(2d_A^{\text{mis}}(L)+(2/3)\epsilon\Delta)}\right)$$
$$\leq 2\exp\left(-\frac{t_2\epsilon^2 n^{1-\alpha}}{4n^{1+\beta}(2+(2/3))}\right)$$
$$= 2n^{-3}.$$

It follows that with probability $1 - O(n^{-3})$, $\frac{1}{t_2}\sum_{i\in[t_2]}Z_i$ is a $(1\pm\epsilon, \pm\epsilon n^{1-\alpha})$-approximation of $d_A^{\text{mis}}(L)$. Consequently, $\tilde{d}_A$ is a $(1\pm\epsilon, \pm\epsilon n^{1-\alpha})$-approximation of $\sum_{u\in V} d_A^{\text{mis}}(u) = 2|E_A^{\text{mis}}|$. The correctness of the lemma follows.

Finally, the space usage of Algorithm 3 is dominated by the sizes of samples $S_1$, $S_2$, and the array $D$. Their total space usage is bounded by $O(t_1 + t_2) = O\left(\frac{1}{\epsilon^2}(n^{1-\beta} + n^{\alpha+\beta})\log n\right)$ words.  $\square$

## 2.2. Description of Algorithm 5 Est-EB

Let $G_B^{\text{mis}} = (V, E_B^{\text{mis}})$ be the subgraph of $G^{\text{mis}}$ that only contains the edges from $E_B^{\text{mis}}$. For each node $u$, we define the cluster of $u$ as $\text{Clu}(u) = \{v \in N(u) \cup \{u\} \mid \text{pivot}(v) = u\}$ if $\text{pivot}(u) = u$, and $\text{Clu}(u) = \emptyset$ otherwise. By Lemma 2.5, if $u \in B$, then we always have $\text{Clu}(u) \subseteq B$. It follows that $B$ is a disjoint union of clusters. We estimate $|E_B^{\text{mis}}|$ by estimating the intra-cluster edges and inter-cluster edges in $E_B^{\text{mis}}$ separately.

For each node $u \in B$, let

$$E_{\text{in}}(u) = \{(v, w) \in E_B^{\text{mis}} \mid v \in \text{Clu}(u) \text{ and } w \in \text{Clu}(u)\},$$
$$E_{\text{out}}(u) = \{(v, w) \in E_B^{\text{mis}} \mid v \in \text{Clu}(u) \text{ and } w \notin \text{Clu}(u)\}.$$

Clearly, $|E_B^{\text{mis}}| = \sum_{u\in B}\left(|E_{\text{in}}(u)| + \frac{1}{2}|E_{\text{out}}(u)|\right)$.

In Algorithm 5, we draw a sample $S$ of size $t = o(n)$ from $B$ (Pass 1) and find all neighbors of the sampled nodes (Pass

**Algorithm 5** Est-EB$(\sigma, \epsilon, \pi, R)$

**Input:** data stream $\sigma = (\sigma_1, \ldots, \sigma_n)$, parameter $\epsilon$, permutation $\pi : V \to [n]$, highest ranked nodes $R$
**Output:** a $(1\pm\epsilon, \pm\epsilon n^{1-\alpha})$-approximation of $|E_B^{\text{mis}}|$
48  $t \leftarrow \frac{8}{\epsilon^2}n^{\alpha+2\beta}\log n$, $n_B \leftarrow 0$
49  Initialize arrays $S[1..t] \leftarrow$ null, $n_{\text{in}}[1..t] \leftarrow 0$, $n_{\text{out}}[1..t] \leftarrow 0$, and $\Gamma[1..t] \leftarrow \emptyset$
   // Pass 1:
50  **foreach** *incoming* $\sigma_j$ **do**
51     |  $p_j \leftarrow$ FindPivot$(\sigma_j, \pi, R)$
52     |  **if** $p_j =$ null **then**
53     |     |  $n_B \leftarrow n_B + 1$
54     |     |  **if** $n_B \leq t$ **then** $S[n_B] \leftarrow \sigma_j$
55     |     |  **else**
56     |     |     |  $i \leftarrow$ random integer from $[n_B]$
57     |     |     |  **if** $i \leq t$ **then** $S[i] \leftarrow \sigma_j$
   // Pass 2:
58  **foreach** *incoming* $\sigma_j$ **do**
59     |  **for** $i \leftarrow 1$ **to** $t$ **do**
60     |     |  **if** $S[i] \sim \sigma_j$ **then** $\Gamma[i] \leftarrow \Gamma[i] \cup \{\sigma_j\}$
   // Pass 3 to Pass $(k+2)$:
61  **do in parallel for each** $i \in [t]$ **and each** $v \in \Gamma[i]$
62     |  $p_v \leftarrow$ PrunedPivot$(\sigma, v, \pi)$
63     |  **if** $p_v \neq S[i]$ **then** $\Gamma[i] \leftarrow \Gamma[i] \setminus \{v\}$
   // Pass $(k+3)$:
64  **foreach** *incoming* $\sigma_j$ **do**
65     |  $p_j \leftarrow$ FindPivot$(\sigma_j, \pi, R)$
66     |  **if** $p_j =$ null **then**
67     |     |  **for** $i \leftarrow 1$ **to** $t$ **do**
68     |     |     |  **foreach** $v \in \Gamma[i]$ **do**
69     |     |     |     |  **if** $\sigma_j \notin \Gamma[i]$ *and* $v \sim \sigma_j$ **then**
70     |     |     |     |     |  $n_{\text{out}}[i] \leftarrow n_{\text{out}}[i] + 1$
71  **for** $i \leftarrow 1$ **to** $t$ **do**
72     |  $n_{\text{in}}[i] \leftarrow |\{(v_1, v_2) \in \Gamma[i] \times \Gamma[i] \mid v_1 \nsim v_2\}|$
73  **return** $\frac{n_B}{t}\sum_{i\in[t]}\left(n_{\text{in}}[i] + \frac{1}{2}n_{\text{out}}[i]\right)$

2). By Lemma 2.6, we know that nodes in $B$ have small degrees, which makes it feasible to store all neighbors of nodes in $S$ in the array $\Gamma[]$. After that, for every $i \in [t]$, we identify the pivot for every node in $\Gamma[i]$ using the streaming version of PrunedPivot, which runs in $k$ passes (Pass 3 to $k + 2$). Next, we discard from each $\Gamma[i]$ all nodes whose pivots are not the $i$-th sample $S[i]$ (Line 63). At this time, for each $i \in [t]$, we have $\Gamma[i] = \text{Clu}(S[i])$ (i.e., $\Gamma[i]$ stores the cluster centered at $S[i]$ if $S[i]$ is a pivot, and $\Gamma[i] = \emptyset$ otherwise). Finally, we use one more pass to compute $n_{\text{in}}[i] (= |E_{\text{in}}(S[i])|)$ and $n_{\text{out}}[i] (= |E_{\text{out}}(S[i])|)$, and consequently estimate the value $|E_B^{\text{mis}}|$.

**Proof of Lemma 2.9.** Due to space constraints, we leave the proof of Lemma 2.9 to Appendix C.1. The proof is similar to that of Lemma 2.8.

## 3. Lower Bounds

In this section, we present two lower bounds demonstrating that both multi-pass data access and additive error are necessary for achieving sublinear space complexity. Our lower bound proofs make use of carefully designed reductions from problems in communication complexity. Due to the space constraints, we leave the detailed proofs to Appendix D.

**Theorem 3.1** (multiple passes are necessary). *For any $c \geq 1$ and $d \in (0, \frac{n}{3}]$, any one-pass 0.49-error $(c, d)$-approximation streaming algorithm for computing the cost of the optimal correlation clustering on an input graph with $n$ nodes in the node-arrival model needs $\Omega(n)$ bits of space.*

**Theorem 3.2** (additive error is needed). *For any $c \geq 1$, any $O(1)$-pass 0.49-error $(c, 0)$-approximation streaming algorithm for computing the cost of the optimal correlation clustering on an input graph with $n$ nodes in the node-arrival model needs $\Omega(n)$ bits of space.*

## 4. Experiments

**Datasets.** We evaluate on three representative datasets: two sparse, highly clustered, real-world graphs and a dense image similarity graph: (1) `Wikipedia`. A hyperlink graph formed from a 2013 snapshot of English Wikipedia (Boldi & Vigna, 2004). The similarity function draws an edge $(\sigma_i, \sigma_j)$ iff page $i$ links to page $j$ or vice versa. It contains $4,206,785$ nodes and $183,879,456$ edges. (2) `LiveJournal`. A social network graph formed from a 2006 snapshot of LiveJournal (Backstrom et al., 2006). The similarity function draws an edge $(\sigma_i, \sigma_j)$ iff user $i$ and user $j$ are friends. It contains $3,997,962$ nodes and $69,362,378$ edges. (3) `ImageNet-21K`. A large image dataset with human-assigned labels (Deng et al., 2009). We use a subset of this dataset, generated by randomly selecting 10% of the labels to include. We then map each image to a vector embedding in $\mathbb{R}^{1024}$ using a pretrained OpenCLIP ViT-g/14 model `laion2b-s34b-b88k` (Ilharco et al., 2021). The similarity function considers cosine similarity between these embeddings. Formally, there is an edge $(\sigma_i, \sigma_j)$ iff $\cos(\text{embedding}(\sigma_i), \text{embedding}(\sigma_j)) > \theta$, for some fixed threshold $\theta$. The resulting graph contains $385,963$ nodes and $110,496,622$ edges.

**Algorithms.** We compare `Pivot`, `PrunedPivot`, `C`[4]`Approx`, `SimpleSampling` (the simple algorithm described in "high-level ideas" in Section 2) on the test datasets, for varying space usage and threshold $k$. We also compare the pivot based algorithm `ASW23-Pivot` and the sparse-dense-decomposition based algorithm `ASW23-SDD` in Assadi et al. (2023), implemented in node-arrival streams. For `C`[4]`Approx`, at Line 21, we just output $(\tilde{m}_A + \tilde{m}_B)$ without the normalization terms.

For `PrunedPivot`, `C`[4]`Approx` and `ASW23-Pivot`, we report the relative error against `Pivot` for the same $\pi$. This metric isolates the error introduced by pruning and sampling, ignoring the inherent variance of `Pivot` itself. The `ASW23-SDD` algorithm is built on a different method (sparse–dense decomposition) and its empirical mean deviates from other algorithms, making it less appropriate to use `Pivot` as a baseline or to report relative error against it. We therefore compare our algorithm `C`[4]`Approx` with `ASW23-SDD` in a separate figure using the absolute clustering cost as the measurement.

**Experimental Environment.** We run our experiments on a Dell PowerEdge R740 server with 2 Intel Xeon Gold 6248R CPUs (totaling 48 cores and 96 threads) and 256 GiB of memory. Our software is implemented in C++ and compiled with Intel oneAPI.

**Space-Accuracy Tradeoff.** For a fixed $k = 15$, we vary the space budget from 2% to 16% and report the empirical mean and standard deviation of the relative error across 100 randomly selected $\pi$.

Figure 1 shows that the mean of `C`[4]`Approx` matches `PrunedPivot` while the standard deviation converges rapidly as space increases, confirming the prediction that the estimator variance decreases with increasing sample size. We note that 10% space is sufficient for `C`[4]`Approx` to get under 2% standard deviation.

Figure 3 compares `C`[4]`Approx`, `SimpleSampling` and `ASW23-Pivot`. We notice that `ASW23-Pivot` and `SimpleSampling` exhibit similar variances: both significantly higher than that of `C`[4]`Approx`. For instance, at 2% space, the standard deviation of `ASW23-Pivot` is about 150% of the mean on Wikipedia and close to 100% on ImageNet, whereas `C`[4]`Approx` achieves roughly 5%. The empirical means of all algorithms are similar, with little to no bias from the mean of `Pivot`.

Figure 4 plots the results of `C`[4]`Approx` and `ASW23-SDD`. It is clear that on both Wikipedia and ImageNet-21K, the standard deviation of `ASW23-SDD` is also much higher than `C`[4]`Approx`. On the Wikipedia dataset, `ASW23-SDD` also has a noticeably higher empirical mean than `C`[4]`Approx`. This is largely due to the sparsity of the graph and the scarcity of high-degree nodes, causing the additive term of `ASW23-SDD` to dominate its output.

**Pass-Accuracy Tradeoff.** We also investigate the trade-off between the accuracy of `C`[4]`Approx` and `PrunedPivot` (compared with `Pivot`) and the number of passes (equal to $k + 4$). For a fixed space budget 4%, we vary $k$ from 2 to 20 and report the empirical mean and standard deviation of the relative error across 100 randomly selected $\pi$.

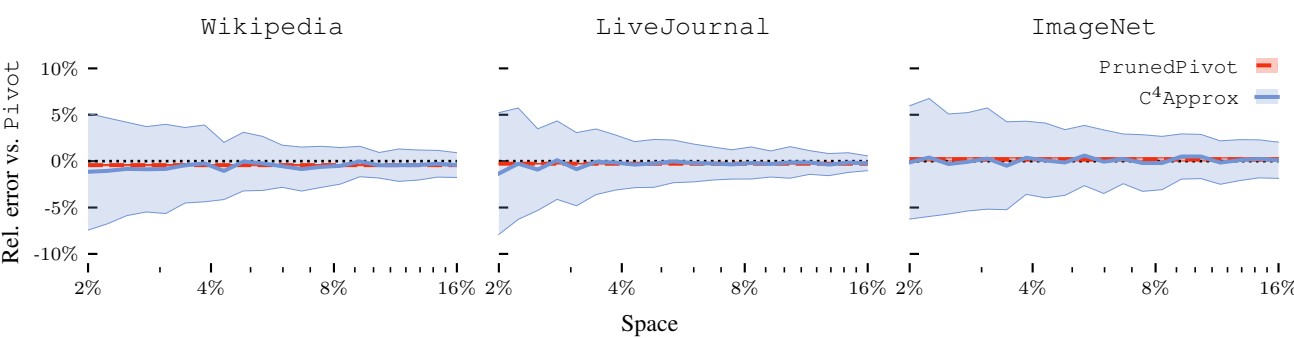

*Figure 1.* Relative error of `PrunedPivot` and `C⁴Approx` w.r.t. `Pivot` for varying space ($k = 15$). Shaded regions show ±1 SD.

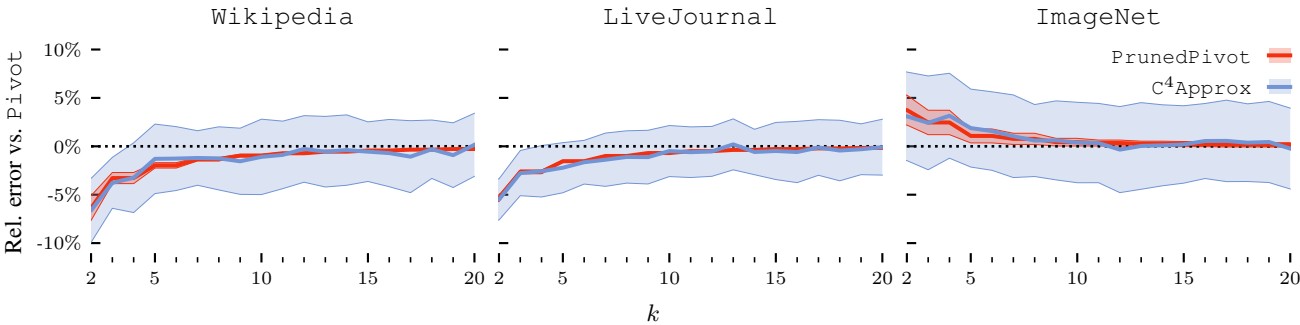

*Figure 2.* Relative error of `PrunedPivot` and `C⁴Approx` w.r.t. `Pivot` for varying $k$ (space fixed at 4%). Shaded regions show ±1 SD.

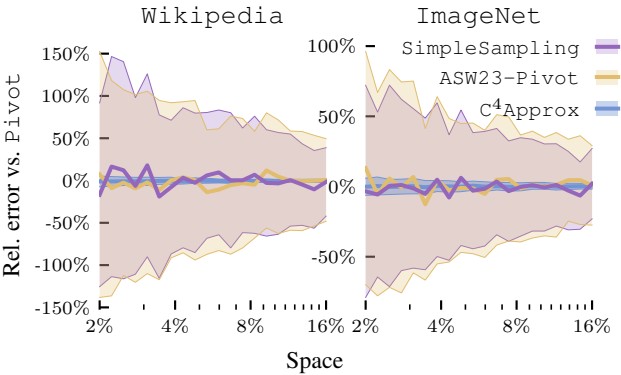

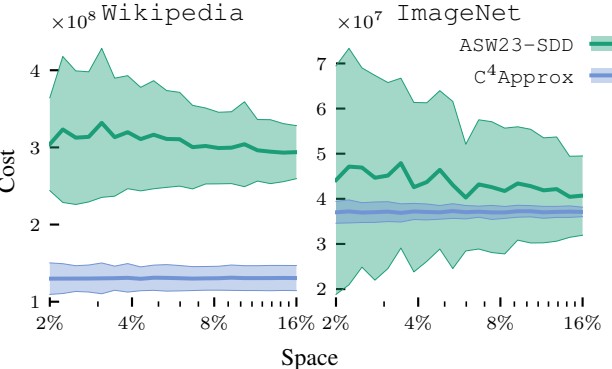

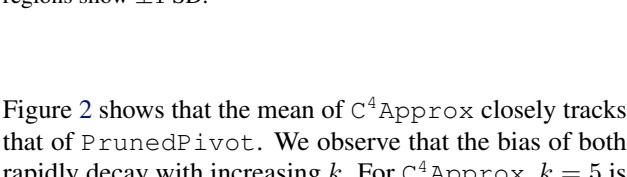

*Figure 3.* `C⁴Approx`, `SimpleSampling`, and `ASW23-Pivot` compared against `Pivot` for varying space and $k = 15$. Shaded regions show ±1 SD.

*Figure 4.* Absolute cost of `C⁴Approx` and `ASW23-SDD` for varying space and $k = 15$. Shaded regions show ±1 SD.

Figure 2 shows that the mean of `C⁴Approx` closely tracks that of `PrunedPivot`. We observe that the bias of both rapidly decay with increasing $k$. For `C⁴Approx`, $k = 5$ is sufficient to get under 2% relative error in expectation.

**Summary.** Our experiments confirm that `C⁴Approx` provides an unbiased, low-variance estimate of `PrunedPivot` in sublinear space and that the accuracy of the estimator improves when either the space or the number of passes is increased. It outperforms all

competitors in terms of accuracy and stability.

## Impact Statement

This paper presents work whose goal is to advance the field of Machine Learning. We do not see any immediate societal consequences of our work.

## Acknowledgment

The work is supported in part by NSF CCF-1844234.

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

---

**Algorithm 6** `PrunedPivot(σ, u, π)`

---

**Input:** data stream $\sigma = (\sigma_1, \ldots, \sigma_n)$, query node $u$, permutation $\pi : V \to [n]$
**Output:** pivot($u$)

74 Initialize a list $L \leftarrow \{(u, u)\}$
75 **for** $\gamma \leftarrow 1$ **to** $k$ **do**
76     // Pass $\gamma$:
77     **foreach** *incoming* $\sigma_j$ **do**
78         **for** $i \leftarrow 1$ **to** $length(L)$ **do**
79             $(x_i, y_i) \leftarrow L[i], (x_{i+1}, y_{i+1}) \leftarrow L[i+1]$
80             **if** $x_i \sim \sigma_j$ and $\pi(x_{i+1}) < \pi(\sigma_j) < \pi(y_i)$ **then**
81                 $L[i] \leftarrow (x_i, \sigma_j)$
82     **while** $(x, y) \leftarrow$ *last element in $L$ and $x = y$* **do**
83         **if** $length(L) \leq 2$ **then return** $x$
84         **else**
85             remove the last two elements in $L$
86     set the last element of $L$ to $(x, x)$
87     add $(y, y)$ as the last element of $L$
88 **return** $u$

---

## A. Math Tools

**Lemma A.1** (Chernoff Bound). *Let $\mathcal{X} = \{x_1, \ldots, x_N\}$ be a finite population of $N$ points and $X_1, \ldots, X_t$ be a random sample drawn without replacement from $\mathcal{X}$. Then for any $\epsilon \in [0, 1]$,*

$$\mathbf{Pr}\left[\left|\frac{1}{t}\sum_{i=1}^{t} X_i - \mu\right| \geq \epsilon\mu\right] \leq 2\exp\left(-\frac{\epsilon^2 \mu t}{3}\right),$$

*where $\mu = \frac{1}{N}\sum_{i=1}^{N} x_i$ is the mean of $\mathcal{X}$.*

**Lemma A.2** (Bernstein Inequality). *With the notations of Lemma A.1, suppose that $x_i \in [0, M]$ for each $i \in [N]$. Then for any $\delta > 0$,*

$$\mathbf{Pr}\left[\left|\frac{1}{t}\sum_{i=1}^{t} X_i - \mu\right| \geq \delta\right] \leq 2\exp\left(-\frac{t\delta^2}{2M\mu + (2/3)M\delta}\right).$$

## B. Streaming Implementation of `PrunedPivot`

Algorithm 6 implements the Pruned Pivot algorithm (Dalirrooyfard et al., 2024) in the streaming model by simulating the first $k$ recursive queries. We maintain the query path in a list $L$. Let $(x_i, y_i)$ be the $i$-th element in $L$, where $x_i$ is the $i$-th node on the query path. During each pass, for each $x_i$, we try to identify $y_i$, the next node to query for $x_i$ after $x_{i+1}$. At the end of each pass, we update the query path in a bottom-up manner.

Let $(x, y)$ be the last element in $L$. Since $x$ is the current query node, if $x \neq y$ (or, $\pi(y) < \pi(x)$), then identifying pivot($x$) requires first identifying pivot($y$). Therefore, we simply add $y$ to the query path and start the next pass. Otherwise, all higher-ranked neighbors of $x$ are not pivots; we thus have pivot($x$) = $x$. Let $v$ be the predecessor of $x$ on the query path. Then $v$ is assigned to the cluster formed by $x$ (i.e. pivot($v$) = $x$). Since both pivot($x$) and pivot($v$) are identified as $x$, if the length of $L$ is at most 2, then we have pivot($u$) = $x$. Otherwise, we remove these two elements from the query path $L$ and repeat updating the query path.

Finally, if after $k$ passes the pivot of $u$ is still not identified, then we can make $u$ a singleton cluster and return $u$, since in this case the query budget has been exhausted. The space usage of Algorithm 6 is bounded by $O(length(L)) = O(k)$.

## C. Missing Proofs in Section 2

### C.1. Proof of Lemma 2.9

We assume that Lemma 2.6 holds, which occurs with probability at least $(1 - n^{-3})$.

For each $i \in [t]$, let $Y_i$ be the $i$-th sample in $S$. Let $Z_i = |B| \cdot (|E_{\text{in}}(Y_i)| + \frac{1}{2}|E_{\text{out}}(Y_i)|)$; we thus have

$$\mathbf{E}[Z_i] = \frac{1}{|B|} \sum_{u \in B} |B| \left( |E_{\text{in}}(u)| + \frac{1}{2}|E_{\text{out}}(u)| \right) = \left| E_B^{\text{mis}} \right|,$$

and

$$Z_i \leq n \left( \frac{1}{2}|\text{Clu}(Y_i)|^2 + \frac{1}{2} \sum_{v \in \text{Clu}(Y_i)} d_v \right) \leq n^{1+2\beta}.$$

Let $\Delta = \max(\left| E_B^{\text{mis}} \right|, n^{1-\alpha})$. By the Bernstein inequality,

$$\mathbf{Pr}\left[ \left| \frac{1}{t} \sum_{i \in [t]} Z_i - \left| E_B^{\text{mis}} \right| \right| \geq \epsilon\Delta \right] \leq 2\exp\left( -\frac{t\epsilon^2\Delta^2}{n^{1+2\beta}(2\left| E_B^{\text{mis}} \right| + (2/3)\epsilon\Delta)} \right)$$

$$\leq 2\exp\left( -\frac{t\epsilon^2 n^{1-\alpha}}{n^{1+2\beta}(2 + (2/3))} \right)$$

$$= 2n^{-3}.$$

Therefore, with probability $1 - O(n^{-3})$, $\tilde{m}_B = \frac{1}{t}\sum_{i \in [t]} Z_i$ is a $(1 \pm \epsilon, \pm\epsilon n^{1-\alpha})$-approximation of $\left| E_B^{\text{mis}} \right|$.

The space usage is dominated by the array of neighboring sets $\Gamma[1..t]$. Since $d_u \leq n^\beta$ for any $u \in B$, the total size of $\Gamma[]$ is bounded by $O(tn^\beta) = O\left( \frac{1}{\epsilon^2}n^{\alpha+3\beta} \log n \right)$. The space needed to find the pivot of each node is bounded by $O(k)$, since we will query at most $k$ nodes. Therefore, the overall space is bounded by $O\left( \frac{k}{\epsilon^2}n^{\alpha+3\beta} \log n \right)$ words.

### C.2. Proof for Lemma 2.10

The pass complexity directly follows from the description of Algorithm 2, Lemma 2.8 and Lemma 2.9. Since we need to store the reference set $R$ before estimating $\left| E_A^{\text{mis}} \right|$ and $\left| E_B^{\text{mis}} \right|$, the overall space complexity is obtained by adding the size of $R$ and the space complexities of Algorithm 3 and Algorithm 5: $O\left( kn^{1-\beta}\log n + \frac{1}{\epsilon^2}(n^{1-\beta} + n^{\alpha+\beta})\log n + \frac{k}{\epsilon^2}n^{\alpha+3\beta}\log n \right) = O\left( \frac{k}{\epsilon^2}(n^{1-\beta} + n^{\alpha+3\beta})\log n \right)$.

We next prove the correctness. Algorithm 2 calls Algorithm 3 and Algorithm 5 with parameter $\epsilon/8$. By Lemma 2.8 and Lemma 2.9, with probability $1 - o(1)$, both $\tilde{m}_A$ and $\tilde{m}_B$ are $(1 \pm (\epsilon/8), \pm(\epsilon/8)n^{1-\alpha})$-approximations of $\left| E_A^{\text{mis}} \right|$ and $\left| E_B^{\text{mis}} \right|$, respectively. Consequently, the output of Algorithm 2 satisfies

$$\frac{\tilde{m}_A + \tilde{m}_B + \frac{3}{8}\epsilon n^{1-\alpha}}{1 - \frac{\epsilon}{8}} \leq \frac{(1 + \frac{\epsilon}{8})\left| E^{\text{mis}} \right| + \frac{5}{8}\epsilon n^{1-\alpha}}{1 - \frac{\epsilon}{8}} \leq \left( 1 + \frac{\epsilon}{3} \right) \left| E^{\text{mis}} \right| + \epsilon n^{1-\alpha},$$

and

$$\frac{\tilde{m}_A + \tilde{m}_B + \frac{3}{8}\epsilon n^{1-\alpha}}{1 - \frac{\epsilon}{8}} \geq \frac{(1 - \frac{\epsilon}{8})\left| E^{\text{mis}} \right| + \frac{1}{8}\epsilon n^{1-\alpha}}{1 - \frac{\epsilon}{8}} \geq \left| E^{\text{mis}} \right|.$$

### C.3. Proof for Theorem 2.11

Let OPT be the cost of the optimal clustering. We first show that Algorithm 2 computes a $(10, n^{1-\alpha})$-approximation of OPT with probability 0.6. Then by $O(1)$ parallel repetitions and taking the median, we can boost the success probability to 0.99.

By Theorem 2.1 (setting $k = 37$), we have with probability at least $\frac{2}{3}$,

$$\mathsf{OPT} \leq \left|E^{\mathrm{mis}}\right| \leq \frac{29}{3}\mathsf{OPT}. \tag{1}$$

Conditioned on (1), by Lemma 2.10 (setting $\epsilon = \frac{1}{10}$), we have with probability $1 - o(1)$,

$$\frac{\tilde{m}_A + \tilde{m}_B + \frac{3}{8}\epsilon n^{1-\alpha}}{1 - \frac{\epsilon}{8}} \leq \left(1 + \frac{\epsilon}{3}\right)\left|E^{\mathrm{mis}}\right| + \epsilon n^{1-\alpha}$$

$$\leq \left(1 + \frac{1}{30}\right)\cdot\frac{29}{3}\cdot\mathsf{OPT} + n^{1-\alpha}$$

$$\leq 10\cdot\mathsf{OPT} + n^{1-\alpha},$$

and

$$\frac{\tilde{m}_A + \tilde{m}_B + \frac{3}{8}\epsilon n^{1-\alpha}}{1 - \frac{\epsilon}{8}} \geq \left|E^{\mathrm{mis}}\right| \geq \mathsf{OPT}.$$

Therefore, Algorithm 2 computes a $\left(10, n^{1-\alpha}\right)$-approximation of $\mathsf{OPT}$ with probability at least $\frac{2}{3} - o(1) \geq 0.6$.

### C.4. Proof of Lemma 2.12

Since $S_1$ are drawn without replacement from $V$, we have $\mathbf{E}[X_u] = \frac{t_1}{n}d_A^{\mathrm{mis}}(u)$.

We start by proving the first item. Note that if $d_A^{\mathrm{mis}}(u) < n^\beta$, then $\mathbf{E}[X_u] < \frac{t_1}{n^{1-\beta}}$. By a Chernoff bound,

$$\mathbf{Pr}\left[X_u \geq \frac{2t_1}{n^{1-\beta}}\right] \leq e^{-\frac{t_1}{3n^{1-\beta}}} \leq n^{-4},$$

which implies that with probability $1 - n^{-4}$, if $X_u \geq \frac{2t_1}{n^{1-\beta}}$, then $d_A^{\mathrm{mis}}(u) \geq n^\beta$.

On the other hand, if $d_A^{\mathrm{mis}}(u) \geq n^\beta$, then $\mathbf{E}[X_u] \geq \frac{t_1}{n^{1-\beta}}$. By another Chernoff bound, we have

$$\mathbf{Pr}[|X_u - \mathbf{E}[X_u]| \geq \epsilon\mathbf{E}[X_u]] \leq 2\exp\left(-\frac{\epsilon^2\mathbf{E}[X_u]}{3}\right) \leq 2\exp\left(-\frac{\epsilon^2 t_1}{3n^{1-\beta}}\right) = 2n^{-4},$$

which implies that when $d_A^{\mathrm{mis}}(u) \geq n^\beta$, $X_u$ is a $(1 \pm \epsilon, 0)$-approximation of $\mathbf{E}[X_u] = \frac{t_1}{n}d_A^{\mathrm{mis}}(u)$ with probability $1 - 2n^{-4}$. By a union bound, with probability $1 - O(n^{-4})$, if $X_u \geq \frac{2t_1}{n^{1-\beta}}$, then $\frac{nX_u}{t_1}$ is a $(1 \pm \epsilon, 0)$-approximation of $d_A^{\mathrm{mis}}(u)$.

For the second item, note that if $d_A^{\mathrm{mis}}(u) > 4n^\beta$, then $\mathbf{E}[X_u] > \frac{4t_1}{n^{1-\beta}}$. By a Chernoff bound,

$$\mathbf{Pr}\left[X_u < \frac{2t_1}{n^{1-\beta}}\right] \leq e^{-\frac{t_1}{3n^{1-\beta}}} \leq n^{-4},$$

which implies that with probability $1 - n^{-4}$, if $X_u < \frac{2t_1}{n^{1-\beta}}$, then $d_A^{\mathrm{mis}}(u) \leq 4n^\beta$.

### C.5. Proof of Lemma 2.13

Let $p_u$ and $p_v$ be the outputs of Algorithm 1 with query nodes $u$ and $v$. We analyze four cases:

1. $p_u = p_v = \texttt{null}$: in this case, $u, v \in B$ and thus $(u, v) \notin E_A^{\mathrm{mis}}$.

2. $p_u \neq \texttt{null}$ *and* $p_v = \texttt{null}$: in this case, $u \in A$ and $v \in B$. By Lemma 2.5, $u$ and $v$ must belong to different clusters. Therefore, $(u, v) \in E_A^{\mathrm{mis}}$ if and only if $u \sim v$, which can be tested given $u$ and $v$.

3. $p_u = \texttt{null}$ *and* $p_v \neq \texttt{null}$: this case is symmetric to the second case.

4. $p_u \neq \texttt{null}$ *and* $p_v \neq \texttt{null}$: in this case, $(u, v) \in E^{\mathrm{mis}}$ if and only if one of the followings holds: (1) $p_u = p_v$ and $u \nsim v$, or (2) $p_u \neq p_v$ and $u \sim v$. Both cases can be easily checked given $u, v$ and $p_u, p_v$.

# D. Lower Bounds

We will make use of the following well-studied problems in communication complexity.

**Definition D.1** (INDEX). *In the* INDEX *problem, Alice holds an $n$-bit vector $x = (x_1, \ldots, x_n)$, and Bob holds an index $b \in [n]$. The goal is for Alice to send a message to Bob that allows him to determine $x_b$.*

**Lemma D.2** (c.f. Kushilevitz & Nisan (1996)). *Any randomized algorithm that solves* INDEX *with success probability at least* $0.51$ *requires* $\Omega(n)$ *bits of communication.*

**Definition D.3** (DISJ). *In the* DISJ *problem, Alice holds an $n$-bit vector $x = (x_1, \ldots, x_n)$, and Bob holds an $n$-bit vector $y = (y_1, \ldots, y_n)$. Their goal is to exchange messages so that they output $1$ if there exists an index $i \in [n]$ with $x_i = y_i = 1$, and $0$ otherwise.*

**Lemma D.4** (Bar-Yossef et al. (2004)). *Any randomized algorithm that solves* DISJ *with success probability at least* $0.51$ *requires* $\Omega(n)$ *bits of communication.*

In our lower bound proofs, we work with points in $\mathbb{R}^d$ under the $\ell_1$ distance, treating each vector as its corresponding point. The similarity function $sim(\cdot, \cdot)$ is defined for two points $p$ and $q$ such that $sim(p, q) = 1$ if $\ell_1(p, q) \leq 1$, and $sim(p, q) = 0$ otherwise.

We denote by $e_i^n$ the $i$-th standard basis vector in $\mathbb{R}^n$, and by $0^n$ the $n$-dimensional zero vector. For vectors $x \in \mathbb{R}^a$ and $y \in \mathbb{R}^b$, we write $x \circ y$ to mean their concatenation, i.e., the vector in $\mathbb{R}^{a+b}$ formed by appending $y$ to $x$.

**Proof of Theorem 3.1.** Let $\ell = \frac{n}{2}$. We perform a reduction from the INDEX problem. Given the input $x = (x_1, \ldots, x_\ell)$, Alice constructs points $(4 \times 1 + x_1) \circ 0^\ell, \ldots (4 \times \ell + x_\ell) \circ 0^\ell$. Given the input $b$, Bob constructs points $(4b) \circ e_1^\ell, \ldots, (4b) \circ e_\ell^\ell$. We note that although each point lies in $\mathbb{R}^{\ell+1}$, they only have at most two non-zero entries, and thus can be represented using $O(1)$ words.

It is not difficult to see that if $\text{INDEX}(x, b) = 1$, then the optimal correlation clustering places each point in its own singleton cluster, incurring zero cost. If $\text{INDEX}(x, b) = 0$, then the optimal correlation clustering also places each point in its own singleton cluster, except that $(4b) \circ 0^\ell$ will be clustered with one of the points in $(4b) \circ e_1^\ell, \ldots, (4b) \circ e_\ell^\ell$. The cost of the clustering is $\ell - 1 > \frac{n}{3}$. Therefore, any one-pass $(c, d)$-approximation streaming algorithm with any $c \geq 1$ and $0 \leq d \leq \frac{n}{3}$ can be used to solve INDEX.

Theorem 3.1 follows directly from Lemma D.2 and the above reduction.

**Proof of Theorem 3.2.** Let $\ell = \frac{n}{4}$. We perform a reduction from the DISJ problem. Given the input $x = (x_1, \ldots, x_\ell)$, Alice constructs $\ell$ points $(2 \times 1) \circ (0 + x_1), \ldots (2 \times \ell) \circ (0 + x_\ell)$ (named by $a_1, \ldots, a_\ell$) and $\ell$ points $(2 \times 1) \circ (-1 + x_1), \ldots (2 \times \ell) \circ (-1 + x_\ell)$ named by $p_1, \ldots, p_\ell$). Given the input $y = (y_1, \ldots, y_\ell)$, Bob constructs $\ell$ points $(2 \times 1) \circ (3 - y_1), \ldots (2 \times \ell) \circ (3 - y_\ell)$ (named by $b_1, \ldots, b_\ell$) and $\ell$ points $(2 \times 1) \circ (4 - y_1), \ldots (2 \times \ell) \circ (4 - y_\ell)$ (named by $q_1, \ldots, q_\ell$). Note that each point lies in $\mathbb{R}^2$.

It is not difficult to see that if $\text{DISJ}(x, y) = 0$, then the optimal correlation clustering places each pair $(a_i, p_i)$ $(i \in [\ell])$ in a cluster and each pair $(b_i, q_i)$ $(i \in [\ell])$ in a cluster, incurring zero cost. If $\text{DISJ}(x, y) = 1$, then the optimal correlation clustering is the same as the case when $\text{DISJ}(x, y) = 0$, but the cost of the clustering becomes $1$. Therefore, $(c, 0)$-approximation streaming algorithm with any $c \geq 1$ can be used to solve DISJ.

Theorem 3.2 follows directly from Lemma D.4 and the above reduction.

