# OpenReview forum: "Estimating Correlation Clustering Cost in Node-Arrival Stream"
_ICML.cc/2026/Conference — ICML 2026 regular_

### Official Review · Reviewer_owSp · 2026-02-25

**Soundness:** 3
**Presentation:** 4
**Significance:** 2
**Originality:** 2
**Overall Recommendation:** 3
**Confidence:** 4

**Summary:**

This paper is about estimating correlation clustering cost of a graph in the streaming model. This the first work on node-streaming model, edge-streaming had been studied before. They give an algorithm that outputs a $(O(1),n^c)$ approximation of the cost of correlation clustering of the graph in $O(1)$ passes. They complement their results by showing that the additive error is needed, and multiple passes are necessary.

**Compliance With Llm Reviewing Policy:**

Affirmed.

**Final Justification:**

I understand the motivations and techniques, however I am still of the opinion that the motivation is not strong, and the techniques are nice but not usable.

**Key Questions For Authors:**

1) In the first motivation example "Quantifying data noise", in the case where there are duplicates the cost of correlation clustering is zero, because the duplicates make cliques. So how measuring the cost identifies the existence of duplicates?
2) I don't understand the second motivation example, can you give an example of how measuring correlation clustering cost can tell you if a similarity function is good?

**Limitations:**

They don't discuss any limitations, but I cannot think of any potential negative societal impact of this work. The limitations mostly go toward the fact that they don't output the clusters (which for the space complexity they are going for is not possible).

**Strengths And Weaknesses:**

strengths:
- The results are well written and seem correct to the extent that I could check.
- The method is clear, they use pruned pivot from previous work that is a modification of the famous pivot algorithm, they tweak it to be adaptable to the streaming setting, and use it as a subroutine. The ideas are stated clearly.
- The experiments are well done and clear.

Weaknesses:
- The main weakness I see is the motivation. I am not convinced that estimating the cost of correlation clustering is an important problem.
- Another more minor weakness is the originality. While I am not of the opinion that every paper should present %100 new ideas, the algorithms are not simple (a lot more complications have beed added to pruned pivot), no major reusable ideas have been introduced and this together with the lack of motivation works negatively for this paper.

Typos/minor:
- theorem 2.1: the result cited is a 3 approximation, you seem to be using markov inequality, can you please clarify that in the manuscript.
- line 298: "the estimate is added to the sum", you don't mention what this sum is representing before this sentence.
- line 314 second column: "if pivot(u) = u ..." this sentence seems to be incomplete/having multiple typos, please fix.

---

> ### Author Rebuttal · Authors · 2026-03-29
>
> We thank the reviewer for the helpful comments.  Please find our response below.
>
> **Motivating Example 1.**
> We realized that "inconsistency" or "ambiguity" is better than "noise" in "quantifying data noise." Noise may cause items corresponding to different ground-truth elements to appear similar, or items corresponding to the same ground-truth element to appear dissimilar. The number of such inconsistent pairs is precisely the correlation clustering cost.
>
> We noticed (after submitting this paper) that a follow-up work of Zhang (2025) introduces the concept of $F_p$-mismatch ambiguity.
>
> - Frequency Moments in Noisy Streaming and Distributed Data under Mismatch Ambiguity, by Kaiwen Liu and Qin Zhang, accepted in PODS 2026
>
> For $p=2$, $F_2$-mismatch ambiguity is essentially the correlation clustering cost. While their work uses this parameter in approximation guarantees, it does not provide streaming algorithms to estimate it. Our work can thus be viewed as complementary, as we focus on approximating this quantity in the data stream model.
>
> **Motivating Example 2.**
> We will clarify that the similarity function depends on a threshold parameter $\theta$. As a concerte example, in our experiments, in the ImageNet-21K dataset, each item is embedded in $\mathbb{R}^{1024}$, and similarity is defined via thresholded cosine similarity. The parameter $\theta$ can be tuned (e.g., by grid search) to minimize the resulting correlation clustering cost.  This is what we meant by using correlation clustering cost as an objective function to learn a good similarity function (or, $\theta$). We will clarify this in the next version.
>
> **Regarding the Novelty of Our Algorithm.**
> We use the PrunedPivot algorithm as a key component of our method; aside from this, we believe the other main ideas are all novel.
>
> **Minor Corrections.**
> - Theorem 2.1: We will clarify the use of Markov's inequality.
> - Line 298: "the estimate is added to the sum" means "the estimate is added to $\tilde{d}_A$".
> - Line 314: The sentence should be "if $pivot(u)=u$, we define the cluster of u as $Clu(u)=$ ... Otherwise, we define $Clu(u)=\emptyset$."

---

> > ### Author Rebuttal · Reviewer_owSp · 2026-04-02
> >
> > I have added my question in a separate comment. I understand the motivation provided for the paper, but I am not convinced that it is important or useful.

---

> > > ### Author Response · Authors · 2026-04-03
> > >
> > > Thank you for your acknowledgment. We cannot see your separate comment. We got your follow-up question "Can you please list your ideas that you think are novel, and if you think they may be usable elsewhere?" from AC.
> > >
> > > Our main novel techniques are: (1) a sublinear-size reference set for node partitioning, and (2) a high-low degree node decomposition for variance reduction. For other problems in node-arrival streams with sublinear memory, one can similarly use a small sketch to identify a set of important nodes and their associated edges for further processing. The high-low degree decomposition is used in our Algorithm 4 to estimate the average degree of a graph, which may serve as a building block for other graph problems.
> > >
> > > Compared to edge-arrival streams, node-arrival streams have received little attention in the literature, even though they are a natural model with many potential applications. We believe that, as more problems are studied in the node-arrival setting, our reference-set based node partitioning and high-low degree decomposition techniques will be useful and may help inspire further work on graph problems in this model.

---

### Official Review · Reviewer_6pb2 · 2026-03-10

**Soundness:** 3
**Presentation:** 3
**Significance:** 3
**Originality:** 3
**Overall Recommendation:** 3
**Confidence:** 3

**Summary:**

This paper studies estimating the optimal correlation clustering cost in the node-arrival streaming model, where the stream consists of raw objects and edges are induced implicitly via a similarity function rather than appearing explicitly in an edge stream. The paper proposes a multi-pass, sublinear-space estimator for approximating the optimum cost, and complements the algorithm with lower bounds indicating that (i) onepass sublinear-space estimation is impossible and (ii) purely multiplicative approximation without additive error is unattainable in sublinear space. Empirically, the method is evaluated on large real datasets and a dense ImageNet-21K similarity graph constructed from OpenCLIP embeddings, with results suggesting competitive accuracy relative to Pivot while storing only a small fraction of nodes.

**Compliance With Llm Reviewing Policy:**

Affirmed.

**Key Questions For Authors:**

Q1: *Oracle cost model*. Do you assume each similarity query is O(1) time and random-access? If the similarity function is implemented via embedding inference or heavy feature comparisons, how should one account for computational and I/O cost in practice?

Q2: *Interpretability of additive error*. When the true OPT cost is small, the additive term may dominate. Can you provide guidance on when to interpret performance using absolute error vs. relative error, and what regimes your estimator is intended for?

Q3: *Sensitivity to theta*. The ImageNet similarity graph depends strongly on the threshold theta and on the embedding backbone. Did you run a theta-sweep or test alternative embedding models to check robustness?

**Limitations:**

The paper’s assumptions are stylized: (i) the “one word per node” storage model abstracts away the real cost
of representing raw objects, and (ii) the similarity oracle is treated as inexpensive, while in practice it can
dominate runtime.

**Strengths And Weaknesses:**

**Strengths**:

S1: *Motivation and model realism*. The node-arrival formulation matches many real pipelines where the data arrive as raw objects and similarity edges are generated on demand, rather than being provided as an edge stream.

S2: *Lower bounds complete the story*. The lower bounds are not just decorative: they concretely justify why multiple passes and additive error are necessary, which aligns well with the design choices in the proposed estimator.

S3: *Large-scale evaluation*. Experiments cover two large sparse graphs (Wikipedia 2013 snapshot and LiveJournal 2006 snapshot) and a dense ImageNet-21K similarity graph (OpenCLIP ViT-g/14 embeddings with a cosine-threshold graph construction). The space–accuracy tradeoff study provides a useful practical picture.

**Weaknesses**:

W1: The “one word per node” abstraction is hard to interpret in practice. The model assumes each node/object can be stored in a single word. For realistic raw objects, this abstraction is far from literal and makes it difficult to translate theoretical memory bounds into actionable engineering guidance.

W2: Multi-pass access is still a strong constraint for streaming deployments. The algorithm relies on multiple passes (parameterized by (k), plus additional passes from subroutines). In many realistic streaming settings, multiple passes over data are expensive; the paper would benefit from a clearer discussion of typical pass counts used in practice and how accuracy degrades when (k) is small.

---

### Official Review · Reviewer_Zbv4 · 2026-03-11

**Soundness:** 3
**Presentation:** 3
**Significance:** 2
**Originality:** 3
**Overall Recommendation:** 4
**Confidence:** 4

**Summary:**

This paper studies the problem of estimating the correlation clustering (CC) cost in the node-arrival streaming model, where the stream consists of vertices and edge labels ($+$ or $-$) are provided by a pairwise function. The CC cost is defined as the minimum number of mismatched edges over all possible vertex partitions.

The main result of the paper is an algorithm in this model that achieves an $(O(1), n^{\kappa})$-approximation using $O(1)$ passes over the stream and $o(n)$ words of space. Furthermore, the authors show that any one-pass algorithm that provides a $(c, n/3)$-approximation requires $\Omega(n)$ bits of space for any $c \ge 1$, and that any $O(1)$-pass $(c,0)$-approximation also requires $\Omega(n)$ bits of space. These results suggest that both \emph{multiple passes} and \emph{additive error} are necessary in order to achieve sublinear-space algorithms for this problem.

It is relatively easy to obtain an $(O(1), \delta n^2)$-approximation in $O(1)$ passes using $poly(\log n)$ space. I suspect that the work of Assadi et al.~(2023) could also yield such a result, possibly even in a single pass. In contrast, this paper shows that the additive error can be reduced to $n^{\kappa}$ (for $\kappa>0$) while still using $o(n)$ space; more specifically, for additive error $n^{\kappa}$, the space usage becomes $n^{1-\kappa/4}$.

The main idea is to construct a partial oracle for $E^{mis}$, the set of mismatched edges produced by \textsc{PrunedPivot}, which originates from Dalirrooyfard et al.~(2024). The oracle is implemented by maintaining a small reference set $R$ in memory consisting of the $o(n)$ highest-ranked vertices according to a random permutation $\pi$. The vertices are then divided into two groups: $A$, whose pivots can be directly determined using $R$, and $B$, whose pivots cannot. The algorithm then treats edges incident to $A$ and edges entirely inside $B$ separately. For example, edges with at least one endpoint in $A$ are handled by estimating the corresponding average degree, while edges with both endpoints in $B$ are handled via sampling from clusters in $B$. Combining these ideas yields the proposed algorithm.

The lower bounds are obtained via reductions from the communication complexity problem \textsc{Index}, which appear to be fairly straightforward.

**Compliance With Llm Reviewing Policy:**

Affirmed.

**Final Justification:**

The paper contains some interesting results, and I maintain a "weak accept".

**Key Questions For Authors:**

Could you provide a more thorough comparison of your results with those of Assadi et al. (2023) in the edge-arrival streaming model, both at the theoretical level and in the experimental evaluation?

**Strengths And Weaknesses:**

-- The paper is generally well written. It adds to the recent line of work on streaming algorithms for correlation clustering, which I consider a natural and fundamental problem.

-- One concern is that the considered node-arrival model is somewhat strong, in the sense that one needs to query a pairwise function to obtain the corresponding edge labels and thus reconstruct the graph. These labels would require $\Omega(n^2)$ space to store explicitly, yet this cost is apparently not counted in the space complexity of the node-arrival streaming model. It would be helpful if the paper could better justify this modeling choice.

-- The proposed approach builds on several previous works, such as \textsc{PrunedPivot} (Dalirrooyfard et al., 2024). Nevertheless, the results appear to be technically non-trivial. In particular, reducing the additive error to $o(n)$ is an interesting improvement. On the other hand, the algorithm requires multiple passes over the stream, which may limit the applicability of the result in practical streaming settings.

-- It would also be helpful to provide a more thorough comparison with the results and techniques of Assadi et al. (2023) in the edge-arrival streaming model.

-- Finally, I am curious why the experimental evaluation does not include a comparison with the algorithm of Assadi et al. (2023).

---

> ### Author Rebuttal · Authors · 2026-03-29
>
> We thank the reviewer for the constructive comments. Our responses are provided below.
>
> **Modeling Choice.**
> Regarding the node-arrival and edge-arrival models, in the node-arrival model, nodes correspond to raw data objects (e.g., images or text) and are revealed only upon arrival. In contrast, the edge-arrival model assumes that all nodes are known in advance. We would like to emphasize that edge labels are not stored explicitly in our algorithm. Instead, they are computed on demand when both endnodes are present in memory (e.g., via cosine similarity between embeddings), incurring no additional storage cost.
>
> We think the node-arrival and edge-arrival models are not directly comparable. In the node-arrival model, edges can only be queried when both endnodes are simultaneously stored, making it impossible to even enumerate all edges in $o(n)$ space within constant passes. In contrast, the edge-arrival model allows a full scan of edges in each pass. On the other hand, the node-arrival model allows querying edges between stored nodes at any time, which may require additional passes in the edge-arrival model.
>
> **Theoretical Comparison with Assadi et al. (2023).**
> After a more careful examination, we realized that the algorithms of Assadi et al. can be implemented in the node-arrival model using two passes (and possibly also one pass). Comparing their algorithms and our proposed algorithm, the key distinction lies in the tradeoff between space and additive error: for the same space bound, their algorithms has significantly larger additive error; equivalently, achieving the same additive error requires significantly more space. Please refer to the first paragraph of Q1 in our response to Reviewer P2Ec for a detailed example.
> Our main contribution is to show that the product of space and additive error can be made subquadratic, which is not achieved by previous work.
>
> From a technical standpoint, the Pivot-based algorithm in Assadi et al. is conceptually similar to the SimpleSampling algorithm in our submission. Although the underlying clusters differ, with ours obtained via PrunedPivot and theirs via LocalPivot, both approaches rely on a small-scale local search to determine the clusters of sampled nodes.
>
> The other algorithm in Assadi et al. is based on a non-pivot clustering approach, namely sparse-dense decomposition, which determines clusters by examining neighborhood overlap among nodes. However, in terms of estimation, the algorithm employs vanilla sampling, resulting in large additive error.
>
> We will add a discussion in our next version.
>
> **Empirical Comparison with Assadi et al. (2023).**
> We implement the SDD-based and Pivot-based algorithms from Assadi et al. (2023), denoted as ASW23-SDD and ASW23-Pivot, with the necessary adaptations to the node-arrival model. We evaluate both methods on two datasets: Wikipedia (a sparse real-world graph) and ImageNet-21K (a dense image similarity graph). We note that ASW23-Pivot is also a pivot-based algorithm, allowing us to include it directly in Figure 3 of the submission, where the Pivot algorithm (Ailon et al.) is used as the ground truth and relative errors of different algorithms are reported. Please see the results in the following [new Figure 1](https://www.dropbox.com/scl/fi/qjn0926t814x8h94kyx9m/space-vs-value-pivot.pdf?rlkey=1kedm0fyxf4xenqa7ebb60ie7&e=1&st=ga12bmfx&dl=0)
>
> The space budget ranges from 2% to 16%, and we report the empirical mean and standard deviation over 100 independent runs for three algorithms, C4Approx, SimpleSampling, and ASW23-Pivot, relative to Pivot. We notice that ASW23-Pivot and SimpleSampling exhibit similar variances, both significantly higher than that of C4Approx. For instance, at 2% space, the standard deviation of ASW23-Pivot is about 150% of the mean on Wikipedia and close to 100% on ImageNet, whereas C4Approx achieves roughly 5%.  The empirical means of all algorithms are similar.
>
> The ASW23-SDD algorithm is built on a different method (sparse–dense decomposition) and its empirical mean deviates from other algorithms, making it less appropriate to use Pivot as a baseline or to report relative error against it. We therefore compare our algorithm C4Approx with ASW23-SDD in a separate figure using the absolute clustering cost as the measurement: [new Figure 2](https://www.dropbox.com/scl/fi/z1u5c3iqb8xwto8zl27c8/space-vs-value-sdd.pdf?rlkey=nwjhsezx6gxaaiuxc6gal8szo&st=61yzp0o1&dl=0)
>
> It is clear that on both Wikipedia and ImageNet-21K, the standard deviation of ASW23-SDD is also much higher than C4Approx. On the Wikipedia dataset, ASW23-SDD also has a noticeably higher empirical mean than C4Approx. This is largely due to the sparsity of the graph and the scarcity of high-degree nodes, causing the additive $\delta n^2$ term to dominate its output.
>
> Overall, these results clearly demonstrate that C4Approx outperforms both ASW23-SDD and ASW23-Pivot, albeit at the cost of requiring more passes.

---

> > ### Author Rebuttal · Reviewer_Zbv4 · 2026-04-03
> >
> > Thanks for your response.

---

### Official Review · Reviewer_P2Ec · 2026-03-12

**Soundness:** 4
**Presentation:** 3
**Significance:** 3
**Originality:** 3
**Overall Recommendation:** 4
**Confidence:** 4

**Summary:**

This paper studies correlation clustering in the node-arrival streaming model, where data objects (not edges) arrive sequentially, and edge labels are derived from a similarity function. The authors present the first streaming algorithm that approximates the cost of correlation clustering using sublinear space in the number of nodes and a constant number of passes. The main result achieves an (O(1), n^{c})-approximation for any constant $c > 0$ using $O(1)$ passes and $o(n)$ space. The paper complements this with lower bounds showing that multiple passes are necessary and that some additive error is unavoidable. Experiments demonstrate that storing only 2% of nodes achieves performance comparable to Pivot and PrunedPivot.

**Compliance With Llm Reviewing Policy:**

Affirmed.

**Final Justification:**

Thanks for the responses.
Will proceed with my initial evaluation.

**Key Questions For Authors:**

1- How does the node-arrival model with a similarity oracle compare to edge-arrival streaming in terms of complexity? Assadi et al. (2023) achieved polylog(n) space for cost estimation in edge-arrival streams, while your algorithm uses $O(n^{(3+\alpha)/4})$ space. Is this gap inherent to the model?

2- The parameter $\alpha$ must be fixed before seeing the stream. Is there an adaptive way to choose α based on stream properties, or is this a fundamental limitation of the approach?

3- Can the optimal cost be estimated directly without going through PrunedPivot? Is the dependence on PrunedPivot's approximation ratio necessary, or an artifact of the analysis?

**Limitations:**

The algorithm provides an $(O(1), n^{1-\alpha})$-approximation, meaning the output is at most $O(1)·OPT + n^{1-\alpha}$. When OPT is small the additive error n^{1-α} dominates entirely, and the estimate provides no meaningful information. The authors do not characterize the regime where their guarantee is useful or discuss how a practitioner would know if they are in this regime.

The final multiplivative approximation ratio is approximately 10, which arises from composing the (9 + 24/(k-1))-approximation of PrunedPivot with the (1 + ε/3) factor from estimation error. The authors do not discuss whether the dependence on PrunedPivot's ratio is inherent to the approach.

Gap with edge-arrival streaming: Is node-arrival fundamentally harder for cost estimation? Can polylog(n) space be achieved with more passes or stronger assumptions?

**Strengths And Weaknesses:**

Strengths
The node-arrival model is natural for applications where raw data objects (images, documents, tweets) arrive as a stream and similarity is computed via a function rather than given explicitly. The two motivating applications, quantifying data noise and testing similarity functions, are concrete and reasonable.
The paper provides an algorithm with sublinear space, proves lower bounds showing that multiple passes are necessary and additive error is unavoidable, and validates experimentally.

Weakness
Several recent papers on correlation clustering in sublinear and streaming models are not cited or discussed:

Cohen-Addad, Lattanzi, Maggiori, and Parotsidis (ICML 2024), "Dynamic Correlation Clustering in Sublinear Update Time," which studies dynamic node streams (vertex additions/deletions) with polylog update time, a closely related model.
They are particularly studying the dynamic node streaming model, which is very relevant. And, a comparison against this paper is needed.

Behnezhad, Cohen-Addad, Charikar, Ghafari, and Ma (ICML 2025), "Correlation Clustering Beyond the Pivot Algorithm," which breaks the 3-approximation barrier using a modified Pivot algorithm directly relevant given this paper's reliance on PrunedPivot.

Cao et al. (STOC 2025), "Solving the Correlation Cluster LP in Sublinear Time," which achieves sublinear-time algorithms via LP techniques.

The main theorem (Theorem 2.11) claims an $(O(1), n^{1-\alpha})$-approximation of the optimal cost, but this guarantee is derived indirectly. The algorithm actually estimates the cost of PrunedPivot's output, which is itself only a $(9 + 24/(k-1))$-approximation of OPT (Theorem 2.1, with probability 2/3). The final multiplicative constant of almost 10 comes from composing these two approximations.

---

> ### Author Rebuttal · Authors · 2026-03-29
>
> We thank the reviewer for the constructive comments. Our responses are provided below.
>
> **Q1.**
> We believe the gap comes from the inherent tradeoff between space complexity and additive error. In the edge-arrival model, Theorem 1 of Assadi et al. (2023) shows that achieving an $(O(1), \delta n^2)$-approximation requires $O(\mathrm{polylog}(n)/\delta^5)$ space, which is polylogarithmic only when $\delta$ is a constant. In contrast, our algorithm achieves an $(O(1), n^{1-\alpha})$-approximation using $O(n^{(3+\alpha)/4}\log n)$ space for any $\alpha \in [0,1)$. For example, setting $\alpha = 0.9$ yields an $(O(1), n^{0.1})$-approximation with still sublinear space. Matching this additive error in Assadi et al. would require setting $\delta = n^{-1.9}$, which in turn gives a space complexity of $O(\mathrm{polylog}(n)/\delta^5) \geq n^{9.5}$, much larger than sublinear.
>
> We think the node-arrival and edge-arrival models are not directly comparable. In the node-arrival model, edges can only be queried when both endnodes are simultaneously stored, making it impossible to even enumerate all edges in $o(n)$ space within constant passes. In contrast, the edge-arrival model allows a full scan of edges in each pass. On the other hand, the node-arrival model permits querying edges between stored nodes at any time, which may require additional passes in the edge-arrival setting.
>
> We would also like to mention that in the node-arrival model, our SimpleSampling algorithm also achieves polylogarithmic space with an $(O(1), \delta n^2)$-approximation.
>
>
> **Q2.**
> The parameter $\alpha$ controls the tradeoff between space and additive error and must be fixed in advance in our algorithm. While it is natural to ask whether $\alpha$ can be selected adaptively, doing so could effectively absorb the additive error term into the $O(1)\cdot \mathrm{OPT}$ term. However, our lower bound states that achieving a purely multiplicative $O(1)$-approximation requires $\Omega(n)$ space, indicating that such adaptivity is generally not possible under sublinear space.
>
>
> **Q3.**
> It is an interesting direction for future work to remove the dependence on the PrunedPivot subroutine. Our approach leverages a key structural property of PrunedPivot: the pivot of a node can be determined via exploration of a constant-size neighborhood. Identifying alternative algorithms with similar locality properties and stronger multiplicative guarantees would be very useful to improve our proposed algorithm.
>
>
> **Related Work**
>
> We thank the reviewer for highlighting these relevant works. We already cite Cao et al. (STOC 2025) in our submission, and will include the other references in the revised version.
>
> - *Cohen-Addad et al. (ICML 2024)* study dynamic correlation clustering under node insertions and deletions, focusing on update time. However, their algorithm requires $O(n \, \mathrm{polylog}(n))$ space, exceeding the sublinear space regime that we are interested.
>
> - *Behnezhad et al. (ICML 2025)* improve upon the classical pivot algorithm by breaking the 3-approximation barrier in a fully dynamic setting, but their approach still requires linear space.
>
> - *Cao et al. (STOC 2025)* propose a sublinear-time algorithm for solving the correlation clustering LP in the centralized setting. Since the LP has exponential size and assumes centralized access, it is unclear how to adapt their approach to the node-arrival streaming model.

---

> > ### Author Rebuttal · Reviewer_P2Ec · 2026-04-02
> >
> > resolved my concerns

---

### Decision · Program_Chairs · 2026-04-30

**Decision:**

Accept (regular)

**Comment:**

Review 6pb2 was disregarded.

This result is about estimating the optimal cost of a correlation clustering instance that arrives node-by-node on a stream. The algorithm uses constantly many passes (which is shown to be unavoidable), a multiplicative-additive approximation (also shown to be unavoidable), and sublinear space. It builds on the recent PrunedPivot work.

The reviewers appreciated the relevance of the node-arrival model (to scenarios where objects arrive on a stream and their similarity is computed using some function), as well as good writing and the strength of the results. It was felt that while the algorithm builds on previous work, the extensions are non-trivial. Experiments were found to be well-done and clear.

Some concerns were raised about the practicality (multiple passes are necessary; and it is not clear that estimating the cost is sufficient or interesting). Some reviewers felt that the newly introduced ideas may be difficult to reuse in future work. The authors have given improved comparisons to related work in the rebuttal; we strongly request that they make these improvements in the manuscript.